# Algorithm selection by rational metareasoning as a model of human strategy selection

**Falk Lieder**
Helen Wills Neuroscience Institute, UC Berkeley
falk.lieder@berkeley.edu

**Dillon Plunkett**
Department of Psychology, UC Berkeley
dillonplunkett@berkeley.edu

**Jessica B. Hamrick**
Department of Psychology, UC Berkeley
jhamrick@berkeley.edu

**Stuart J. Russell**
EECS Department, UC Berkeley
russell@cs.berkeley.edu

**Nicholas J. Hay**
EECS Department, UC Berkeley
nickjhay@berkeley.edu

**Thomas L. Griffiths**
Department of Psychology, UC Berkeley
tom_griffiths@berkeley.edu

## Abstract

Selecting the right algorithm is an important problem in computer science, because the algorithm often has to exploit the structure of the input to be efficient. The human mind faces the same challenge. Therefore, solutions to the algorithm selection problem can inspire models of human strategy selection and vice versa. Here, we view the algorithm selection problem as a special case of metareasoning and derive a solution that outperforms existing methods in sorting algorithm selection. We apply our theory to model how people choose between cognitive strategies and test its prediction in a behavioral experiment. We find that people quickly learn to adaptively choose between cognitive strategies. People's choices in our experiment are consistent with our model but inconsistent with previous theories of human strategy selection. Rational metareasoning appears to be a promising framework for reverse-engineering how people choose among cognitive strategies and translating the results into better solutions to the algorithm selection problem.

## 1 Introduction

To solve complex problems in real-time, intelligent agents have to make efficient use of their finite computational resources. Although there are general purpose algorithms, particular problems can often be solved more efficiently by specialized algorithms. The human mind can take advantage of this fact: People appear to have a toolbox of cognitive strategies [1] from which they choose adaptively [2, 3]. How these choices are made is an important, open question in cognitive science [4]. At an abstract level, choosing a cognitive strategy is equivalent to the algorithm selection problem in computer science [5]: given a set of possible inputs $\mathcal{I}$, a set of possible algorithms $\mathcal{A}$, and a performance metric, find the selection mapping from $\mathcal{I}$ to $\mathcal{A}$ that maximizes the expected performance. Here, we draw on a theoretical framework from artificial intelligence–*rational metareasoning* [6]–and Bayesian machine learning to develop a mathematical theory of how people should choose between cognitive strategies and test its predictions in a behavioral experiment.

In the first section, we apply rational metareasoning to the algorithm selection problem and derive how the optimal algorithm selection mapping can be efficiently approximated by model-based learning when a small number of features is predictive of the algorithm's runtime and accuracy. In Section 2, we evaluate the performance of our solution against state-of-the-art methods for sorting

algorithm selection. In Sections 3 and 4, we apply our theory to cognitive modeling and report a behavioral experiment demonstrating that people quickly learn to adaptively choose between cognitive strategies in a manner predicted by our model but inconsistent with previous theories. We conclude with future directions at the interface of psychology and artificial intelligence.

## 2 Algorithm selection by rational metareasoning

Metareasoning is the problem of deciding which computations to perform given a problem and a computational architecture [6]. Algorithm selection is a special case of metareasoning in which the choice is limited to a few sequences of computations that generate complete results. According to rational metareasoning [6], the optimal solution maximizes the *value of computation* (VOC). The VOC is the expected utility of acting after having performed the computation (and additional computations) minus the expected utility of acting immediately. In the general case, determining the VOC requires solving a Markov decision problem [7]. Yet, in the special case of algorithm selection, the hard problem of planning which computations to perform how often and in which order reduces to the simpler one-shot choice between a small number algorithms. We can therefore use the following approximation to the VOC from [6] as the performance metric to be maximized:

$$\text{VOC}(a; \mathbf{i}) \approx \mathbb{E}_{P(S|a,\mathbf{i})}[S] - \mathbb{E}_{P(T|a,\mathbf{i})}[\text{TC}(T)] \tag{1}$$

$$m(\mathbf{i}) = \arg\max_{a \in \mathcal{A}} \text{VOC}(a; \mathbf{i}), \tag{2}$$

where $a \in \mathcal{A}$ is one of the available algorithms, $\mathbf{i} \in \mathcal{I}$ is the input, $S$ and $T$ are the score and runtime of algorithm $a$ on input $\mathbf{i}$, and $\text{TC}(T)$ is the opportunity cost of running the algorithm for $T$ units of time. The score $S$ can be binary (correct vs. incorrect output) or numeric (e.g., error penalty). The selection mapping $m$ defined in Equation 2 depends on the conditional distributions of score and runtime ($P(S|a, \mathbf{i})$ and $P(T|a, \mathbf{i})$). These distributions are generally unknown, but they can be learned. Learning an approximation to the VOC from experience, i.e. *meta-level learning* [6], is a hard technical challenge [8], but it is tractable in the special case of algorithm selection.

Learning the conditional distributions of score and runtime separately for every possible input is generally intractable. However, in many domains the inputs are structured and can be approximately represented by a small number of features. Concretely, the effect of the input on score and runtime is mediated by its features $\mathbf{f} = (f_1(\mathbf{i}), \cdots, f_N(\mathbf{i}))$:

$$P(S|a, \mathbf{i}) = P(S|\mathbf{f}, a) = P(S|f_1(\mathbf{i}), \cdots, f_N(\mathbf{i}), a) \tag{3}$$

$$P(T|a, \mathbf{i}) = P(T|\mathbf{f}, a) = P(T|f_1(\mathbf{i}), \cdots, f_N(\mathbf{i}), a). \tag{4}$$

If the features are observable and the distributions $P(S|f_1(\mathbf{i}), \cdots, f_N(\mathbf{i}), a)$ and $P(T|f_1(\mathbf{i}), \cdots, f_N(\mathbf{i}), a)$ have been learned, then one can very efficiently compute an estimate of the expected value of applying the algorithm to a novel input. To learn the distributions $P(S|f_1(\mathbf{i}), \cdots, f_N(\mathbf{i}), a)$ and $P(T|f_1(\mathbf{i}), \cdots, f_N(\mathbf{i}), a)$ from examples, we assume simple parametric forms for these distributions and estimate their parameters from the scores and runtimes of the algorithms on previous problem instances.

As a first approximation, we assume that the runtime of an algorithm on problems with features $\mathbf{f}$ is normally distributed with mean $\mu(\mathbf{f}; a)$ and standard deviation $\sigma(\mathbf{f}; a)$. We further assumed that the mean is a 2$^{\text{nd}}$ order polynomial in the extended features $\tilde{\mathbf{f}} = (f_1(\mathbf{i}), \cdots, f_N(\mathbf{i}), \log(f_1(\mathbf{i})), \cdots, \log(f_N(\mathbf{i})))$ and that the variance is independent of the mean:

$$P(T|\mathbf{f}; a, \alpha) = \mathcal{N}(\mu_T(\mathbf{f}; a, \alpha), \sigma_T(a)) \tag{5}$$

$$\mu_T(\mathbf{f}; a, \alpha) = \sum_{k_1=0}^{2} \cdots \sum_{k_N=0}^{2-\sum_{i=1}^{N-1} k_i} \alpha_{k_1, \cdots, k_N; a} \cdot \tilde{f}_1^{k_1} \cdot \ldots \cdot \tilde{f}_N^{k_N} \tag{6}$$

$$P(\sigma_T(a)) = \text{Gamma}(\sigma_T^{-1}; 0.01, 0.01), \tag{7}$$

where $\alpha$ are the regression coefficients. Similarly, we model the probability that the algorithm returns the correct answer by a logistic function of a second order polynomial of the extended features:

$$P(S = 1|a, \mathbf{f}, \beta) = \frac{1}{1 + \exp\left(\sum_{k_1=0}^{2} \cdots \sum_{k_N=0}^{2-\sum_{i=1}^{N-1} k_i} \beta_{k_1, \cdots, k_N; a} \cdot \tilde{f}_1^{k_1} \cdot \ldots \cdot \tilde{f}_N^{k_N}\right)}, \tag{8}$$

with regression coefficients $\beta$. The conditional distribution of a continuous score can be modeled analogously to Equation 5, and we use $\gamma$ to denote its regression coefficients.

If the time cost is a linear function of the algorithm's runtime, i.e. $\text{TC}(t) = c \cdot t$ for some constant $c$, then the value of applying the algorithm depends only on the expectations of the runtime and score distributions. For linear scores

$$\mathbb{E}_{P(S,T|a,\mathbf{i})}\left[S - \text{TC}(T)\right] = \mu_S(\mathbf{f}(\mathbf{i}); a, \gamma) - c \cdot \mu_T(\mathbf{f}(\mathbf{i}); a, \alpha), \tag{9}$$

and for binary scores

$$\mathbb{E}_{P(S,T|a,\mathbf{i})}\left[S - \text{TC}(T)\right] = \mathbb{E}_{P(\beta|s,a,\mathbf{i})}\left[P(S = 1; \mathbf{i}, \beta)\right] - c \cdot \mu_T(f(\mathbf{i}); a, \alpha). \tag{10}$$

We approximated $\mathbb{E}_{P(\beta|s,a,\mathbf{i})}\left[P(S = 1; \mathbf{i}, \beta)\right]$ according to Equation 10 in [9].

Thus, the algorithm selection mapping $m$ can be learned by estimating the parameters $\alpha$ and $\beta$ or $\gamma$. Our method estimates $\alpha$ by Bayesian linear regression. When the score is binary, $\beta$ is estimated by variational Bayesian logistic regression [9], and when the score is continuous, $\gamma$ is estimated by Bayesian linear regression. For Bayesian linear regression, we use conjugate Gaussian priors with mean zero and unit variance, so that the posterior distributions can be computed very efficiently by analytic update equations. Given the posterior distributions on the parameters, we compute the expected VOC by marginalization. When the score is continuous $\mu_S(\mathbf{f}(\mathbf{i}); a, \gamma)$ is linear in $\gamma$ and $\mu_T(\mathbf{f}(\mathbf{i}); a, \alpha)$ is linear in $\alpha$. Thus integrating out $\alpha$ and $\gamma$ with respect to the posterior yields

$$\text{VOC}(a; \mathbf{i}) = \mu_S\left(f(\mathbf{i}); a, \mu_{\gamma|\mathbf{i},s}\right) - c \cdot \mu_T\left(f(\mathbf{i}); a, \mu_{\alpha|\mathbf{i},t}\right), \tag{11}$$

where $\mu_\alpha$ and $\mu_\gamma$ are posterior means of $\alpha$ and $\gamma$ respectively. This implies the following simple solution to the algorithm selection problem:

$$a(i; c) = \arg\max_{a \in \mathcal{A}}\left(\mu_S(\mathbf{f}(\mathbf{i}); a, \mu_{\gamma|\mathbf{i}_{\text{train}}, s_{\text{train}}}) - c \cdot \mu_T(\mathbf{f}(\mathbf{i}); a, \mu_{\alpha|\mathbf{i}_{\text{train}}, t_{\text{train}}})\right). \tag{12}$$

For binary scores, the runtime component is predicted in exactly the same way, and a variational approximation to the posterior predictive density can be used for the score component [9].

To discover the best model of an algorithm's runtime and score, our method performs feature selection by Bayesian model choice [10]. We consider all possible combinations of the regressors defined above. To efficiently find the optimal set of features in this exponential large model space, we exploit that all models are nested within the full model. This allows us to efficiently compute Bayes factors using Savage-Dickey ratios [11].

# 3   Performance evaluation against methods for selecting sorting algorithms

Our goal was to evaluate rational metareasoning not only against existing methods but also against human performance. To facilitate the comparison with how people choose between cognitive strategies, we chose to evaluate our method in the domain of sorting. Algorithm selection is relevant to sorting, because there are many sorting algorithms with very different characteristics. In sorting, the input $\mathbf{i}$ is the sequence to be sorted. Conventional sorting algorithms are guaranteed to return the elements in correct order. Thus, the critical difference between them is in their runtimes, and runtime depends primarily on the number of elements to be sorted and their presortedness. The number of elements determines the relative importance of the coefficients of low (e.g., constant and linear) versus high order terms (e.g., $n^2$, or $n \cdot \log(n)$) whose weights differ between algorithms. Presortedness is important because it determines the relative performance of algorithms that exploit pre-existing order, e.g., insertion sort, versus algorithms that do not, e.g., quicksort.

According to recent reviews [12, 13], there are two key methods for sorting algorithm selection: Guo's decision-tree method [14] and Lagoudakis et al.'s recursive algorithm selection method [15]. We thus evaluated the performance of rational metareasoning against these two approaches.

## 3.1   Evaluation against Guo's method

Guo's method learns a decision-tree, i.e. a sequence of logical rules that are applied to the list's features to determine the sorting algorithm [14]. Guo's method and our method represent inputs by

| test set | performance | 95% CI | Guo's performance | p-value |
|---|---|---|---|---|
| $D_{\text{sort5}}$ | 99.78% | $[99.7\%, 99.9\%]$ | 98.5% | $p < 10^{-15}$ |
| nearly sorted lists | 99.99% | $[99.3\%, 100\%]$ | 99.4% | $p < 10^{-15}$ |
| inversely sorted lists | 83.37% | $[82.7\%, 84.1\%]$ | 77.0% | $p < 10^{-15}$ |
| random permutations | 99.99% | $[99.2\%, 100\%]$ | 85.3% | $p < 10^{-15}$ |

Table 1: Evaluation of rational metareasoning against Guo's method. Performance was measured by the percentage of problems for which the method chose the fastest algorithm.

the same pair of features: $f_1 = |\mathbf{i}|$, the length of the list to be sorted, and $f_2$, a measure of presortedness. Concretely, $f_2$ estimates the number of inversions from the number of runs in the sequence, i.e. $f_2 = \frac{f_1}{2} \cdot \text{RUNS}(\mathbf{i})$, where $\text{RUNS}(\mathbf{i}) = |\{m : i_m > i_{m+1}\}|$. This measure of presortedness can be computed much more efficiently than the number of inversions.

Our method learns the conditional distributions of runtime and score given these two features, and uses them to approximate the conditional distributions given the input (Equations 3–4). We verified that our method can learn how runtime depends on sequence length and presortedness (data not shown). Next, we subjected our method to Guo's performance evaluation [14]. We thus evaluated rational metareasoning on the problem of choosing between insertion sort, shell sort, heapsort, merge sort, and quicksort. We matched our training sets to Guo's $D_{\text{Sort4}}$ in the number of lists (i.e. 1875) and the distributions of length and presortedness. We provided the run-time of all algorithms rather than the index of the fastest algorithm. Otherwise, the training sets were equivalent. For each of Guo's four test sets, we trained and evaluated rational metareasoning on 100 randomly generated pairs of training and test sets. The first test set mimicked Guo's $D_{sort5}$ problem set [14]. It comprised 1000 permutations of the numbers 1 to 1000. Of the 1000 sequences, 950 were random permutations and 50 were nearly-sorted. The nearly-sorted lists were created by applying 10 random pair-wise permutations to the numbers 1–1000. The sequences contained between 1 and 520 runs (mean=260, SD=110). The second test set comprised 1000 nearly-sorted lists of length 1000. Each list was created by applying 10 different random pair-wise permutations to the numbers 1 to 1000. The third test set comprised 100 lists in reverse order. The fourth test set comprised 1000 random permutations.

Table 1 compares how frequently rational metareasoning chose the best algorithm on each test set to the results reported by Guo [14]. We estimated our method's expected performance $\theta$ by its average performance and 95% credible intervals. Credible intervals (CI) were computed by Bayesian inference with a uniform prior, and they comprise the values with highest posterior density whose total probability is 0.95. In brief, rational metareasoning significantly outperformed Guo's decision-tree method on all four test sets. The performance gain was highest on random permutations: rational metareasoning chose the best algorithm 99.99% rather than only 85.3% of the time.

## 3.2 Evaluation against Lagoudakis et al.'s method

Depending on a list's length Lagoudakis et al.'s method chooses either insertion sort, merge sort, or quicksort [15]. If merge sort or quicksort is chosen the same decision rule is applied to each of the two sublists it creates. The selection mapping from lengths to algorithms is determined by minimizing the expected runtime [15]. We evaluated rational metareasoning against Lagoudakis et al.'s recursive method on 21 versions of Guo's $D_{\text{sort5}}$ test set [14] with $0\%, 5\%, \cdots, 100\%$ nearly-sorted sequences. To accommodate differences in implementation and architecture, we recomputed Lagoudakis et al.'s solution for the runtimes measured on our system. Rational metareasoning chose between the five algorithms used by Guo and was trained on Guo's $D_{\text{sort4}}$ [14]. We compare the performance of the two methods in terms of their runtime, because none of the numerous choices of recursive algorithm selection corresponds to our method's algorithm choice.

On average, our implementation of Lagoudakis et al.'s method took $102.5 \pm 0.83$ seconds to sort the 21 test sets, whereas rational metareasoning finished in only $27.96 \pm 0.02$ seconds. Rational metareasoning was thus significantly faster ($p < 10^{-15}$). Next, we restricted the sorting algorithms available to rational metareasoning to those used by Lagoudakis et al.'s method. The runtime increased to $47.90 \pm 0.02$ seconds, but rational metareasoning remained significantly faster than Lagoudakis

et al.'s method ($p < 10^{-15}$). These comparisons highlight two advantages of our method: i) it can exploit presortedness, and ii) it can be used with arbitrarily many algorithms of any kind.

### 3.3 Discussion

Rational metareasoning outperformed two state-of-the-art methods for sorting algorithm selection. Our results in the domain of sorting should be interpreted as a lower bound on the performance gain that rational metareasoning can achieve on harder problems such as combinatorial optimization, planning, and search, where the runtimes of different algorithms are more variable [12]. Future research might explore the application of our theory to these harder problems, take into account heavy-tailed runtime distributions, use better representations, and incorporate active learning.

Our results show that rational metareasoning is not just theoretically sound, but it is also competitive. We can therefore use it as a normative model of human strategy selection learning.

## 4   Rational metareasoning as a model of human strategy selection

Most previous theories of how humans learn when to use which cognitive strategy assume basic model-free reinforcement learning [16–18]. The REinforcement Learning among Cognitive Strategies model (RELACS [17]) and the Strategy Selection Learning model (SSL [18]) each postulate that people learn just one number for each cognitive strategy: the expected reward of applying it to an unknown problem and the sum of past rewards, respectively. These theories therefore predict that people cannot learn to instantly adapt their strategy to the characteristics of a new problem. By contrast, the Strategy Choice And Discovery Simulation (SCADS [16]) postulates that people separately learn about a strategy's performance on particular types of problems and its overall performance and integrate the resulting predictions by multiplication.

Our theory makes critically different assumptions about the mental representation of problems and each strategy's performance than the three previous psychological theories. First, rational metareasoning assumes that problems are represented by multiple features that can be continuous or binary. Second, rational metareasoning postulates that people maintain separate representations of a strategy's execution time and the quality of its solution. Third, rational metareasoning can discover non-additive interactions between features. Furthermore, rational metareasoning postulates that learning, prediction, and strategy choice are more rational than previously modeled. Since our model formalizes substantially different assumptions about mental representation and information processing, determining which theory best explains human behavior will teach us more about how the human brain represents and solves strategy selection problems.

To understand when and how the predictions of our theory differ from the predictions of the three existing psychological theories, we performed computer simulations of how people would choose between sorting strategies. In order to apply the psychological theories to the selection among sorting strategies, we had to define the reward ($r$). We considered three notions of reward: i) correctness ($r \in \{-0.1, +0.1\}$; these numbers are based on the SCADS model [16]), ii) correctness minus time cost ($r - c \cdot t$, where $t$ is the execution time and $c$ is a constant), and iii) reward rate ($r/t$). We evaluated all nine combinations of the three theories with the three notions of reward. We provided the SCADS model with reasonable problem types: short lists (length $\leq 16$), long lists (length $\geq 32$), nearly-sorted lists (less than $10\%$ inversions), and random lists (more than $25\%$ inversions). We evaluated the performance of these nine models against the rational metareasoning in the selection between seven sorting algorithms: insertion sort, selection sort, bubble sort, shell sort, heapsort, merge sort, and quicksort. To do so, we trained each model on 1000 randomly generated lists, fixed the learned parameters and evaluated how many lists each model could sort per second. Training and test lists were generated by sampling. Sequence lengths were sampled from a Uniform($\{2, \cdots, u\}$) distribution where $u$ was 10, 100, 1000, or 10000 with equal probability. The fraction of inversions between subsequent numbers was drawn from a Beta($2, 1$) distribution. We performed 100 train-and-test episodes. Sorting time was measured by selection time plus execution time. We estimated the expected sorting speed for each model by averaging. We found that while rational metareasoning achieved $88.1 \pm 0.7\%$ of the highest possible sorting speed, none of the nine alternative models achieved more than $30\%$ of the maximal sorting speed. Thus, the time invested in metareasoning was more than offset by the time saved with the chosen strategy.

# 5 How do people choose cognitive strategies?

Given that rational metareasoning outperformed the nine psychological models in strategy selection, we asked whether the mind is more adaptive than those theories assume. To answer this question, we designed an experiment for which rational metareasoning predicts distinctly different choices.

## 5.1 Pilot studies and simulations

To design an experiment that could distinguish between our competing hypotheses, we ran two pilot studies measuring the execution time characteristics of cocktail sort (CS) respectively merge sort (MS). For each pilot study we recruited 100 participants on Amazon Mechanical Turk. In the first pilot study, the interface shown in Figure 1(a) required participants to follow the step-by-step instructions of the cocktail sort algorithm. In the second pilot study, participants had to execute merge sort with the computer interface shown in Figure 1(b). We measured their sorting times for lists of varying length and presortedness. Then, based on this data, we estimated how long comparisons and moves take using each strategy. This led to the following sorting time models:

$$\mathrm{T_{CS}} = \hat{t}_{\mathrm{CS}} + \varepsilon_{\mathrm{CS}}, \hat{t}_{\mathrm{CS}} = 19.59 + 0.19 \cdot n_{\mathrm{comparisons}} + 0.31 \cdot n_{\mathrm{moves}}, \varepsilon_{\mathrm{CS}} \sim \mathcal{N}(0, 0.21 \cdot \hat{t}_{\mathrm{CS}}^2) \quad (13)$$

$$\mathrm{T_{MS}} = \hat{t}_{\mathrm{MS}} + \varepsilon_{\mathrm{MS}}, \hat{t}_{\mathrm{MS}} = 13.98 + 1.10 \cdot n_{\mathrm{comparisons}} + 0.52 \cdot n_{\mathrm{moves}}, \varepsilon_{\mathrm{MS}} \sim \mathcal{N}(0, 0.15 \cdot \hat{t}_{\mathrm{MS}}^2) \quad (14)$$

We then used these sorting time models to simulate 104 candidate strategy selection experiments according to each of the 10 models. We found several potential experiments for which rational metareasoning makes qualitatively different predictions than all of the alternative psychological theories, and we chose the one that achieved the best compromise between discriminability and duration.

According to the two runtime models (Equations 13–14) and how many comparisons and moves each algorithm would perform, people should choose merge sort for long and nearly inversely sorted sequences and cocktail sort for sequences that are either nearly-sorted or short. For the chosen experimental design, the three existing psychological theories predicted that people would fail to learn this contingency; see Figure 2. By contrast, rational metareasoning predicted that adaptive strategy selection would be evident from the choices of more than 70% of our participants. Therefore, the chosen experimental design was well suited to discriminate rational metareasoning from previous theories. The next section describes the strategy choice experiment in detail.

## 5.2 Methods

The experiment was run online[1] with 100 participants recruited on Amazon Mechanical Turk and it paid $1.25. The experiment comprised three stages: training, choice, and execution. In the *training stage*, each participant was taught to sort lists of numbers by executing the two contrasting strategies tested in the pilot studies: cocktail sort and merge sort. On each of the 11 training trials, the participant was instructed which strategy to use. The interface enforced that he or she correctly performed each step of that strategy. The interfaces were the same as in the pilot studies (see Figure 1). For both strategies, the chosen lists comprised nearly reversely sorted lists of length 4, 8, and 16 and nearly-sorted lists of length 16 and 32. For the cocktail sort strategy, each participant was also trained on a nearly inversely sorted list with 32 elements. Participants first practiced cocktail sort for five trials and then practiced merge sort. The last two trials contrasted the two strategies on long, nearly-sorted sequences with identical length. Nearly-sorted lists were created by inserting a randomly selected element at a different random location of an ascending list. Nearly inversely sorted lists were created applying the same procedure to a descending list. In the *choice phase*, participants were shown 18 test lists. For each list, they were asked to choose which sorting strategy they would use, if they had to sort this sequence. Participants were told that they would have to sort one randomly selected list with the strategy they chose for it. The test lists comprised six instances of each of three kinds of sequences: long and nearly inversely sorted, long and nearly-sorted, and short and nearly-sorted. The order of these sequences was randomized across participants. In the *execution phase*, one of the 12 short lists was randomly selected, and the participant had to sort it using the strategy he or she had previously chosen for that list.

To derive theoretical predictions, we gave each model the same information as our participants.

Figure 1: Interfaces used to train participants to perform (a) cocktail sort and (b) merge sort in the behavioral experiment.

## 5.3 Results

Our participants took $24.7 \pm 6.7$ minutes to complete the experiment (mean $\pm$ standard deviation). The median number of errors per training sequence was $2.45$, and $95\%$ of our participants made between $0.73$ and $12.55$ errors per training sequence. In the choice phase, $83\%$ of our participants were more likely to choose merge sort when it was the superior strategy (compared to trials when it was not). We can thus be $95\%$ confident that the population frequency of this adaptive strategy choice pattern lies between $74.9\%$ and $89.4\%$; see Figure 2b). This adaptive choice pattern was significantly more frequent than could be expected, if strategy choice was independent of the lists' features ($p < 10^{-11}$). This is consistent with our model's predictions but inconsistent with the predictions of the RELACS, SSL, and SCADS models. Only rational metareasoning correctly predicted that the frequency of the adaptive strategy choice pattern would be above chance ($p < 10^{-5}$ for our model and $p \geq 0.46$ for all other models). Figure 2(b) compares the proportion of participants exhibiting this pattern with the models' predictions. The non-overlapping credible intervals suggest that we can be $95\%$ confident that the choices of people and rational metareasoning are more adaptive than those predicted by the three previous theories (all $p < 0.001$). Yet we can also be $95\%$ confident that, at least in our experiment, people choose their strategy even more adaptively than rational metareasoning ($p \leq 0.02$).

On average, our participants chose merge sort for $4.9$ of the $6$ long and nearly inversely sorted sequences ($81.67\%$ of the time, $95\%$ credible interval: $[77.8\%; 93.0\%]$), but for only $1.79$ of the $6$ nearly-sorted long sequences ($29.83\%$ of the time, $95\%$ credible interval: $[12.9\%, 32.4\%]$), and for only $1.62$ of the $6$ nearly-sorted short sequences ($27.00\%$ of the time, $95\%$ credible interval: $[16.7\%, 40.4\%]$); see Figure 2(a). Thus, when merge sort was superior, our participants chose it significantly more often than cocktail sort ($p < 10^{-10}$). But, when merge sort was inferior, they chose cocktail sort more often than merge sort ($p < 10^{-7}$).

## 5.4 Discussion

We evaluated our rational metareasoning model of human strategy selection against nine models instantiating three psychological theories. While those nine models completely failed to predict our participants' adaptive strategy choices, the predictions of rational metareasoning were qualitatively correct, and its choices came close to human performance. The RELACS and the SSL model failed, because they do not represent problem features and do not learn about how those features affect each strategy's performance. The model-free learning assumed by SSL and RELACS was maladaptive because cocktail sort was faster for most training sequences, but was substantially slower for the

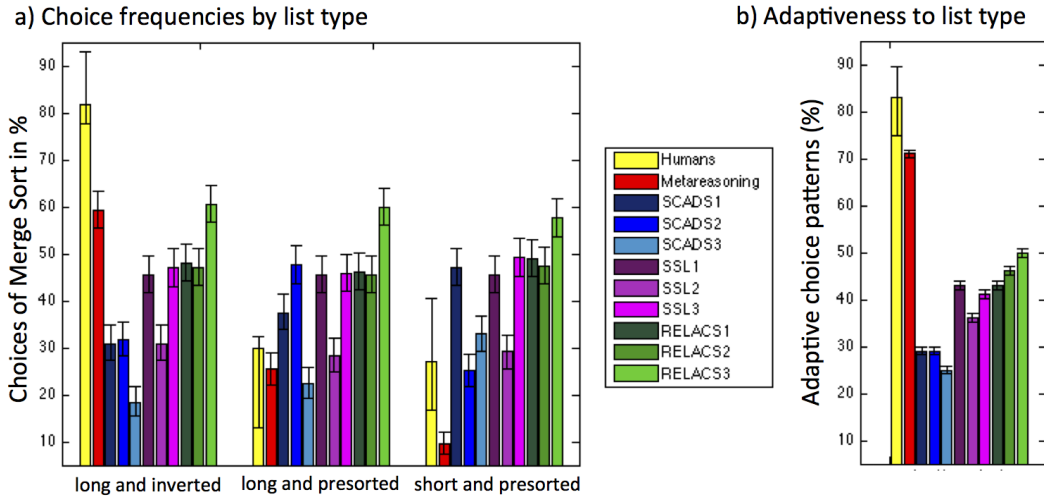

a) Choice frequencies by list type

b) Adaptiveness to list type

Figure 2: Pattern of strategy choices: (a) Relative frequency with which humans and models chose merge sort by list type. (b) Percentage of participants who chose merge sort more often when it was superior than when it was not. Error bars indicate $95\%$ credible intervals.

long, nearly inversely sorted test sequences. The SCADS model failed mainly because its suboptimal learning mechanism was fooled by the slight imbalance between the training examples for cocktail sort and merge sort, but also because it can neither extrapolate nor capture the non-additive interaction between length and presortedness. Instead human-like adaptive strategy selection can be achieved by learning to predict each strategy's execution time and accuracy given features of the problem. To further elucidate the human mind's strategy selection learning algorithm, future research will evaluate our theory against an instance-based learning model [19].

Our participants outperformed the RELACS, SSL, and SCADS models, as well as rational metareasoning in our strategy selection task. This suggests that neither psychology nor AI can yet fully account for people's adaptive strategy selection. People's superior performance could be enabled by a more powerful representation of the sequences, perhaps one that includes reverse-sortedness, or the ability to choose strategies based on mental simulations of their execution on the presented list. These are just two of many possibilities and more experiments are needed to unravel people's superior performance. In contrast to the sorting strategies in our experiment, most cognitive strategies operate on internal representations. However, there are two reasons to expect our conclusions to transfer: First, the metacognitive principles of strategy selection might be domain general. Second, the strategies people use to order things mentally might be based on their sorting strategies in the same way in which mental arithmetic is based on calculating with fingers or on paper.

## 6   Conclusions

Since neither psychology nor AI can yet fully account for people's adaptive strategy selection, further research into how people learn to select cognitive strategies may yield not only a better understanding of human intelligence, but also better solutions to the algorithm selection problem in computer science and artificial intelligence. Our results suggest that reasoning about which strategy to use might contribute to people's adaptive intelligence and can save more time than it takes. Since our framework is very general, it can be applied to strategy selection in all areas of human cognition including judgment and decision-making [1, 3], as well as to the discovery of novel strategies [2]. Future research will investigate human strategy selection learning in more ecological domains such as mental arithmetic, decision-making, and problem solving where people have to trade off speed versus accuracy. In conclusion, rational metareasoning is a promising theoretical framework for reverse-engineering people's capacity for adaptive strategy selection.

**Acknowledgments.** This work was supported by ONR MURI N00014-13-1-0341.

## Footnotes

[1]http://cocosci.berkeley.edu/mturk/falk/StrategyChoice/consent.html

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
