[Reviews · NeurIPS 2014]

Submitted by Assigned_Reviewer_1

Summary: This very strong paper proposes a rational model for algorithm selection based on problem features and Bayesian regression. The model is shown to be effective computationally and to better predict human performance than comparable models.

This paper is the epitome of a strong NIPS paper. The paper is clearly written and addresses an interesting problem. There is both a nice computational result about the algorithm and a cognitive model that is tested with a brief experiment.

My only substantive concern is that too much is made of the rational component of the approach. A lot of the success (both computationally and as a cognitive model) comes from breaking the problem space into the features rather than the rational inference per se. It would be nice to see this approach pitted against an RL model with function approximation, instead of the simple RL model considered.

Minor points:

-Fig 1 has tiny fonts.
-line 38: “by specialized algorithms” no a.
-Equations 11-12 were a bit hard to follow and certainly did not seem simple
-One of the strongest parts of the paper is the very serious comparison against other algorithms that was undertaken. That is not sufficient usual and should be commended.
Summary: This very strong paper proposes a rational model for algorithm selection based on problem features and Bayesian regression. The model is shown to be effective computationally and to better predict human performance than comparable models.

Submitted by Assigned_Reviewer_25

This paper is refreshing in its scope and ambition; I've reviewed and read many NIPS papers that are just one tiny idea explored in excruciating detail. However, despite the paper's considerable breadth and the effort placed into evaluation, I feel this paper may walk the line too far to the side of broad and shallow, versus narrow and deep. I lay out specific concerns below that limit the impact of the results presented.

The area of algorithm selection / metareasoning is key in both AI and cog sci.
I am surprised that more work hasn't been done in this important area.
(I don't know the literature, but I am trusting the authors' review which
mostly consists of decade-old work. They do fail to cite work in cognitive
architectures like SOAR and ACT-R which probably doesn't have a lot to
offer in terms of how people learn metastrategies.)

I am confused by the authors' decision to focus on the domain of sorting
algorthms because this domain doesn't lend itself well to the rational
metareasoning (RM) model. RM attempts to optimize a combined measure of
algorithm accuracy and algorithm time/opportunity cost. However, for sorting
algorithms, don't all algorithms achieve perfect accuracy? And as a result,
isn't the modeling of scoring (Equations 3, 8, 9, and 10) irrelevant?

I also wonder whether there is much benefit of a Bayesian linear regression
approach over simple ridge regression, given that only parameter means are
being utilized (Equation 12)? That said, the representation selected seems
very sensible (using the two features, their logs, and 2nd order polynomial
terms involving these features and their logs).

The work claims to contribute both to state-of-the-art AI approaches to
algorithm selection and to the psychological theory of metareasoning. I have
questions about its contributions to each, which I'll address separately.

With regard to AI, the authors compare their results to two existing models.
The Lagoudakis method doesn't appear to utilize presortedness. As a result,
it seems more like a straw man than a serious contender. With regard to
the comparison to Guo, I had some concern that (based on the text on line 160)
a different measure of presortedness was being used by Guo, but the authors
assure me in their rebuttal that they use the same representation as Guo.

With regard to the work's contribution to cognitive science: In human strategy
selection, the trade off between accuracy and opportunity cost is key, and the
opportunity cost involved is not only the run time of an algorithm, but the
cost of selecting an algorithm, as reflected in Simon's notion of satisficing.
Thus, I question selection of sorting algorithms as the most suitable domain
for studying human strategy selection. Although the experimental set up
in Section 5 is elaborate and impressive, the coarse performance statistics
(proportion of merge sort selection and overall quality of selection)
hardly make a compelling argument that the RM model is correct. All we
know from the experiment is that both the RM model and people are pretty
good at selecting strategies, whereas the other models are not. This result
gives us little insight as to whether the RM model is a good cognitive model.

I couldn't find much detail about training the models used in Section 5, but if
these are really meant to be cognitive models they should be trained on the
same data that people had available during practice, and the same total number
of trials. The author rebuttal assures me that the training data is identical.

MINOR COMMENTS:

[093]: A Gaussian distribution may be acceptable for sorting algorithm
runtimes, but it's probably not the best choice for modeling human response
times. On a range of tasks from simple button presses to memory retrieval to
complex problem solving, reaction times tend to have a long-tailed asymmetric
distribution.

[091]: The standard deviation does not appear to be a polynomial in the
extended features (Equation 7)

[168]: The table caption should explain the performance measure. I believe
it is the percentage of runs in which the optimal sorting algorithm was
selected.

[295]: The text says that Equations 13 and 14 suggest the conditions
under which merge sort will be chosen over cocktail sort. However, the
coefficients on n_comparisons and n_moves in Equation 13 are both smaller
than the corresponding coefficients in Equation 14, so it seems to me
that cocktail sort should be chosen for all but the very shortest lists.
Summary: The work addresses an important challenge to AI and to cognitive science. The authors try a straightforward approach to learning strategy selection and get sensible results in a limited domain (selecting a sorting algorithm).

Submitted by Assigned_Reviewer_32

The authors propose to model the selection of the appropriate algorithm for a task as a metacognitive process by maximizing the value of computation, defined as the amount of expected utility of acting after having carried out a computation that exceeds the expected utility without carrying out the computation. The authors approximate this value by learning to predict the score and the runtime of an algorithm on the basis of features, which are chosen by Bayesian feature selection. Applying this meta-reasoning to the problem of choosing the algorithm for minimizing the duration of sorting lists shows that the proposed computation performs better than two previously proposed methods, one based on decision trees and one based on recursive algorithm selection using dynamic programming.
Finally, the authors present a set of behavioral experiments in which human subjects had to sort lists according to either cocktail sort or merge sort by following respective instructions. After estimating from behavioral data the time to carry out individual sorting actions according to the two algorithms, the authors applied this model to the case in which subjects chose the algorithm to sort new sequences. The human behavior was better predicted by the metacognitive strategy compared to previous models of how humans choose to sort.

This is one of the best papers of the last three years I have been reviewing for Nips. The only recommendation I would like to give the authors for improving the manuscript is to provide more details for the reader on:
- feature selection by Bayesian model choice,
- allows us to efficiently compute Bayes factors,
- Lagoudakis et al.’s method,
- an explanation for why Lagoudakis et al.’s method performs so much worse.

P6:
“to an descending list.”
a
Summary: Great manuscript formalizing how to choose one algorithm for sorting applied to machine sorting and to human performance.
Author Feedback
Author rebuttal: Thank you very much for your positive feedback and constructive criticism! The changes suggested by Assigned_Reviewer1 and Assigned_Reviewer32 and the minor comment of Assigned_Reviewer_25 are very sensible, easily implemented, and will make our paper clearer. To address Assigned_Reviewer_1's concern that we overstate the contribution of rational learning to our model's success at predicting our participants' decisions, we will carefully rephrase our conclusion in lines 403-406 so that it attributes less of our cognitive model's success to its rationality assumption and more to its representations. We believe that Assigned_Reviewer_25's more severe concerns about our work's potential impact on AI and cognitive science are based on misunderstandings that can perhaps be avoided by adding and rephrasing individual sentences. We go through these concerns, address them, and correct underlying misunderstandings in the following three paragraphs address the reviewer's concerns about our paper's potential impact on artificial intelligence, its potential impact on cognitive science, and additional technical questions respectively.

Assigned_Reviewer_25 questioned the relevance of our contribution to artificial intelligence, because "line 160 seems to imply that Guo's work uses a different measure of presortedness." However, this interpretation is incorrect. To the contrary, the presortedness measures were exactly the same. We will rewrite the sentence in line 160 to avoid the misleading description and explicitly state that we used the presortedness measure that Guo had chosen for the reason mentioned in line 160.

Assigned_Reviewer_25 also questioned the relevance of our contribution to cognitive science for four reasons. First, the reviewer points out that the benefit of metareasoning has to be evaluated against its time cost. We have done this in the evaluation reported in the last paragraph of section 4. We found that despite the time invested in metareasoning about strategies, the rational metareasoning model accomplished the task significantly faster than alternative models that don't perform metareasoning. Second, Assigned_Reviewer_25 points out that while we can rule out the three existing psychological theories (RELACS, SSL, and SCADS) and found that our model's predictions were consistent with people's behavior, the brain could achieve the performance predicted by rational metareasoning in different ways that are still unknown. This is true, and we will tone down our discussion in lines 403-406 to more accurately reflect what we can and what we cannot conclude from our results. However, the criticized limitation is a general problem of all cognitive models ever tested rather than a specific weakness of our paper. As Assigned_Reviewer_1 pointed out, our comparative evaluation of 10 different cognitive models already goes beyond standard practice in cognitive science, and we will follow up this initial evaluation with testing more fine-grained predictions of rational metareasoning in future work. Third, Assigned_Reviewer_25 argues that our models "should be trained on the same data that people had available during practice." This is exactly what we did in the simulations reported in section 5. We will add a sentence to section 5.2 to make this explicit. Fourth, Assigned_Reviewer_25 is concerned that this would give the rational metareasoning model an unfair advantage "because it is much simpler with fewer parameters to constrain based on the relatively small number of training trials". This concern is unjustified, because the opposite is true: The representation of rational metareasoning uses up to 13 parameters per strategy whereas the competing psychological models represent each strategy's performance by only one (RELACS and SSL) or two parameters (SCADS).

Assigned_Reviewer_25 wondered why we chose sorting even though the sorting algorithms we considered are guaranteed to return a correct solution. The answer is that we were looking for a behavioral experiment in which we could control and directly observe our participants' strategies and relate them to well-understood algorithms from computer science. Thus Assigned_Reviewer_25 is correct that the reported evaluations did not exploit our method's predictive model of the score (Equations 3,8,9,10) but only from its predictive model of the runtime. Yet, it outperformed existing methods and cognitive models nevertheless. Furthermore, we will evaluate our method's accuracy prediction capacity in future applications to problem solving and decision-making. The reviewer also wondered whether there is any benefit of Bayesian regression of ridge regression in our model. We implicitly answer this question in the last paragraph of section 2. In brief, our probabilistic model allows for efficient automatic model-order selection. Furthermore, the additional cost of Bayesian regression is negligible, because we have analytic update equations for all parameters.

We are thus confident that minor changes to the text will be sufficient to address the reviewers' concerns.